# Low Vertebrobasilar Velocity Is Associated with a Higher Risk of Posterior Circulation Ischemic Lesions

**DOI:** 10.3390/jcm11051396

**Published:** 2022-03-03

**Authors:** Po-Chi Chan, Hua-Si Huang, Kuan-Wen Chen, Hsin-Yi Chi

**Affiliations:** 1Department of Neurology, Show Chwan Memorial Hospital, Chunghua 50008, Taiwan; bear514@outlook.com; 2Department of Radiology, Show Chwan Memorial Hospital, Chunghua 50008, Taiwan; cshe190@csh.org.tw; 3Department of Neurology, Chung Shan Medical University Hospital, Taichung 40201, Taiwan; emchenily@gmail.com; 4School of Medicine, Chung Shan Medical University, Taichung 40201, Taiwan

**Keywords:** vertebrobasilar insufficiency, brainstem stroke, central vertigo, doppler sonography

## Abstract

Background: Transcranial color-coded sonography (TCCS) is used as a real-time tool to evaluate patients suspected of having vertebrobasilar insufficiency (VBI). However, the sonographic criteria for VBI remain inconclusive. The purpose of this study was to analyze the velocity in the vertebrobasilar system, which links the risk for posterior circulation infarction (POCI) and total ischemic stroke (TIS) in patients with VBI. Methods: Patients’ data were retrospectively reviewed if they were suspected of having VBI within a 2-year period. Baseline characteristics, brain images, and a series of sonography data were recorded and analyzed. We compared vertebrobasilar (VB) velocities in different age groups and in patients with infarctions. Results: A total of 875 patients were enrolled, with 112 and 427 candidates in the POCI and TIS groups, respectively. The mean velocity (MV)s of BA and bilateral VAs were all negatively correlated with age (all *p* < 0.001). The adjusted odds ratio was 2.55 (1.58–4.13, *p* < 0.001) in POCI and 1.75 (1.15–2.67, *p* = 0.009) in TIS if the mean velocity of the VB arteries was below 15 cm/s. Conclusions: Low VB velocity detected in TCCS was more commonly associated with ageing-related changes and a higher risk of both POCI and TIS. Recognition and aggressive treatment for these patients are necessary.

## 1. Introduction

Dizziness is the most frequent symptom among patients with posterior circulation infarction (POCI) [1,2]; it is also a typical warning symptom for vertebrobasilar ischemia or vertebrobasilar insufficiency (VBI) [3,4,5]. VBI has been identified since 1945 as a syndrome of intermittent transient ischemic attacks involving posterior circulation [2,6]. The spells of ischemia result from insufficient blood flow to the brain, either in the vertebral or basilar arteries [4]. The advent of newer technology in the past few decades has made it possible to more effectively investigate patients with vertebrobasilar (VB) territory disease [2,7]. However, recognizing high-risk patients with nonspecific symptoms, such as dizziness, is a challenge. Subtle discomforts may progress to basilar arterial occlusion, causing more severe functional impairments such as quadriparesis or coma [5,8].

Sonography, with its widespread availability and real-time hemodynamics, is an ideal tool for clinical screening [9]. The detection rate of vascular disorders by sonography is as valuable as computed tomography (CT) or magnetic resonance imaging (MRI) [10,11]. Neurologists use sonography as an important screening tool for assessing stroke risk factors [12]. The main pathophysiology of ischemic spells in the posterior circulation is hypoperfusion [2]; therefore, the VB system is a target for sonography evaluation [13]. However, the sonographic criteria for hypoperfusion in the VB system remain inconclusive [2]. Thus, we aimed to correlate VB velocity in sonography with the prevalence of cerebral vascular events and evaluated the risk factors of VB hypoperfusion.

## 2. Materials and Methods

This retrospective study was approved by the Institutional Review Board of Show Chwan Memorial Hospital, Chunghua, Republic of China. Informed consent was obtained from all subjects according to the World Medical Association Declaration of Helsinki.

### 2.1. Candidates

From January 2014 to October 2015, we reviewed 1018 consecutive outpatients and inpatients who were suspected of having VBI in the Neurology Department of the Changhua Show Chwan Memorial Hospital. Clinical suspicion of VBI was based on either one of the following criteria: 1. dizziness, excluding peripheral vertigo; or 2. dizziness, with focal neurological symptoms. Patients with incomplete sonography data or without brain imaging data were excluded from this study. Medical and laboratory data of enrolled patients were recorded, including age, sex, systemic diseases, renal function, lipid profile, drug use, electrocardiography data, and history of previous stroke or transient ischemic attack.

### 2.2. Sonography

All patients underwent a transcranial color-coded sonography using the IE-33 system (Philips Medical System, Bothell, WA, USA). This was conducted by experienced technicians. The target of the blood flow exam was the V4 segment of the vertebral artery (VA) and proximal portion of the basilar artery (BA). The angle between the ultrasound beam and the direction of blood flow was adjusted at the technicians’ discretion.

Ultrasonographic data for analysis, including mean velocity (MV), peak systolic velocity (PSV), end diastolic velocity (EDV), and pulsatility index (PI), were recorded for bilateral VAs and BA. The MVs for the BA and VAs were grouped together in 5 cm/s increments, ranging from ≤15 to ≥60 cm/s.

### 2.3. Brain Images and Criteria of Stroke

All patients underwent brain CT or MRI. More than 70% of the candidates in our study came from outpatient clinics, and most of the patients received brain CT only. In our study, those who underwent brain MRI were mostly inpatients.

All image findings were reviewed and discussed by radiologists and neurologists. Lacune lesions with clear margins and containing small CSF cavies were recognized as scattered or isolated distributions. The lesions with the following characteristics were diagnosed as white matter hyperintensity: ill-defined margins, patchy, or diffuse appearance in subcortical area, halos, or multiple punctate lesions. Any infarcts in the brainstem, cerebellum, occipital, medial temporal lobes, and thalamus were recorded as POCI; any infarcts in locations other than the abovementioned areas were recorded as anterior circulation infarction (ACI). Total ischemic stroke (TIS) was defined as the sum of posterior and anterior infarction.

### 2.4. Statistics

Statistical analyses were performed with SPSS software (version 17.0; SPSS Inc., Chicago, IL, USA). Continuous data are expressed as mean ± SD. Normality of continuous data was tested using the Kolmogorov–Smirnov method. The correlations between MV and age, and PI and age for the different arteries were determined using Pearson correlation coefficients.

The differences in MV between POCI, ACI, and TIS were evaluated. We identified and determined the cutoff value of MV using the odds ratio. A multinomial logistic regression was performed to analyze stroke risk among the different groups, expressed as adjusted odds ratios (AORs; adjusted for age, sex, hypertension, and diabetes mellitus), and 95% confidence intervals (CIs). Significant differences were defined as *p* < 0.05. Regression diagnostics and Bonferroni correction were applied in checking the variance inflation factor (VIF) values and *p*-values adjusted for multiplicity.

## 3. Results

A total of 875 patients (458 men; mean age 68.77 ± 12.56 years, ranging from 25 to 97 years) were included in this study. We found that patients with clinically diagnosed VBI were mainly older individuals (Table 1). Of these, 48.8% had ischemic brain lesions. Not all patients with clinically diagnosed VBI had vascular lesions in the posterior circulation. Conversely, most lesions were located in the anterior circulation. In addition, some patients had both ACI and POCI (6.4%). Traditional stroke-related risk factors were associated differently in POCI and TIS groups. Common risk factors in these two groups included sex, diabetes mellitus, hypertension, and history of dyslipidemia (statin drug use). In sonographic findings, significant results fell on the MV and PI value in the left VA.

We found that the MVs of BA and bilateral VAs were all negatively correlated with age (Table 2); there was a positive correlation between PI and age in all three vessels.

In Table 3, the AORs and 95% CIs of the grouped MVs are summarized. After adjusting for age, sex, hypertension, and diabetes mellitus, there were increased AORs for TIS and POCI (all *p* values ≤ 0.01) in groups of velocity less than 15 or 20 cm/s. In the MV group > 60 cm/s, higher AORs of TIS were noted. From the results of Table 3, we determined that lower VB velocities occurred at a higher rate in POCI patients than in other patients.

## 4. Discussion

Low vertebrobasilar velocity was associated with ageing and a higher risk of POCI in this study. According to our experience, clinical diagnosis of VBI is more common in elderly patients, which was reaffirmed by the age distribution in this study. Ageing-related VB system changes include both decreased velocity and increased arteriosclerosis [14] (Table 2). This evidence suggests that vascular ageing in the VB system makes the elderly vulnerable to VBI.

The sonographic criteria of VBI remain inconclusive, which limits the clinical application and diagnostic rate of VBI using sonography [9,15,16,17]. Traditionally, Spencer’s curve theory is applied to explain the hemodynamic model of vascular stenosis and flow velocity [18,19]. According to this theory, the majority of vascular disorders have a high cutoff value for velocity. The unique anatomical structure of the VB arteries, which originate from two small arteries and connect to a larger one, are different from the internal carotid artery system. Therefore, Spencer’s curve theory might need to be adjusted for this condition [20,21]. Series studies suggest that either unilateral or bilateral vertebral hypoplasia is associated with a higher risk of POCI or VBI [22,23]. Although the diagnostic criteria for vertebral hypoplasia are based on extracranial sonography data [13], the lower velocity of the hypoplasia artery also suggests poor collateral supplementation and subsequently results in VBI or POCI [24]. In this study, increased AORs of POCI and TIS were noted in the MV cutoff values below 15 or MV 20 cm/s (*p* value < 0.01). For clinical application, if the patients’ data fall in this category, they might need to be carefully evaluated because of the relative increased risk of TIS and POCI.

High velocities as the cutoff value revealed some contrary results in our study. High PSV of the BA or VA has been considered an important diagnostic criterion for VB system abnormalities, especially in cases of vascular stenosis [7,17,19,25]. However, in this study, groups with velocities higher than 55 or 60 cm/s were not closely related to POCI. This has several possible explanations. First, most of our patients were older and their VB blood flow velocity had decreased to low levels [14]. Therefore, using a high velocity criterion for VBI diagnosis might result in inadequate patient inclusion and statistical values. From the perspective of pathophysiology, in the group with high velocity, it might indicate some structural abnormality; compensatory high velocity may ensure adequate blood supply to brain tissues [18,21]; however, in groups with low velocity, it usually means a poor perfusion reservoir and higher risk for ischemia [4,19]. This might explain why there were not as many incidences of stroke in the high-velocity group compared to the low-velocity group.

In our study, 12.8% of patients fit the diagnosis of POCI and 42.4% had infarcts in the anterior circulation. These data revealed the difficulty in clinically diagnosing VBI due to the overlapping of ischemic symptoms and signs in these two vascular territories [26]. We found that half of these POCI patients (56/112) had co-morbidities of ACI. This theory was also confirmed in previous studies [27]. The ageing- and atherosclerosis-related VB system infarction might indicate diffused vascular disorders in whole intracranial arteries [28]. Following this point of view, the clinical application of sonography data in the VB system may offer referential values for the anterior circulation systems. Owing to demographic differences, the elderly and women have higher failure rates of performing sonography over the temporal window [29]. However, there is no such limitation in the occipital window. Based on this study, abnormalities in VB flow in older people might suggest similar abnormalities over the other intracranial system.

There were some limitations to our study. The limitations of TCCS include the lack of continuous access to the technology [30]; thus, we could not collect the optimal mean flow velocity range that corresponded with VB hypoperfusion. Meanwhile, the precision of VB velocity using sonography is operator-dependent and it is difficult to reduce technique errors during examinations. Most of our POCIs were confirmed by MRI (89.3%).If we only included the patients who received a brain MRI study, then there would be selection bias in the patient group distribution. Since more than 70% of the candidates in our study came from an outpatient clinic, they received only a brain CT study. However, we cannot deny there is the possibility that POCI lesions were omitted from patients who received CT only. We had incomplete data on previous cardiovascular events, smoking, and alcohol use; therefore, we chose not to analyze these data and may have missed these important risk factors.

## 5. Conclusions

Our study findings revealed that lower VB velocity may be associated with higher rates of both anterior and posterior infarctions. The findings also confirm the importance of ageing-related velocity and vascular changes in the VB system. Further evaluation and aggressive treatment for high-risk patients identified by sonographic data should be considered.

## Figures and Tables

**Table 1 jcm-11-01396-t001:** Characteristics of patients.

Parameter	Total Patients (*n* = 875)	POCI (*n* = 112)	TIS (*n* = 427)
Sex	Male 458/Female 417	Male 70/Female 42 *p* value:0.026	Male 266/Female 161 *p* value: <0.001
Age (years)	68.77 ± 12.56 (25–97)	69.58 ± 11.43 *p* value: 0.466	71.05 ± 11.65 *p* value: <0.001
Diabetes mellitus	39.43%	52.68% *p* value: 0.003	43.22% *p* value:0.026
Hypertension	68.22%	85.71% *p* value: <0.001	81.49% *p* value: <0.001
Dyslipidemia	45.37%	49.11% *p* value:0.417	49.18% *p* value: 0.026
CKD (eGFR < 50)	17.71%	24.11% *p* value:0.064	23.89% *p* value: <0.001
Af	9.94%	12.5% *p* value:0.314	12.6% *p* value: 0.009
Statin drug use	33.48%	42.85% *p* value: 0.032	40.75% *p* value: <0.001
MV and PI of BA	33.33 ± 14.34 cm/s, 1.1 ± 0.3	31.51 ± 15.14 cm/s, 1.16 ± 0.36 *p* value: 0.15; *p* value:0.13	32.61 ± 16.06 cm/s, 1.14 ± 0.32 *p* value: 0.148; *p* value: 0.001
MV and PI of RT VA	20.79 ± 13.11 cm/s, 1.1 ± 0.3	26.03 ± 13.85 cm/s, 1.19 ± 0.36 *p* value: 0.98; *p* value: 0.42	26.62 ± 14.52 cm/s, 1.19 ± 0.35 *p* value: 0.04; *p* value: <0.001
MV and PI of LT VA	28.47 ± 11.75 cm/s, 1.1 ± 0.3	25/14 ± 13.63 cm/s, 1.19 ± 0.38 *p* value: 0.001; *p* value: 0.001	27.43 ± 12.75 cm/s, 1.13 ± 0.33 *p* value: 0.011; *p* value: 0.01
Brain image MRI diagnosed stroke		CT: 53.1%; MRI: 46.9% POCI: 89.3%; ACI 63.9%	
TIS rate/ACI rate/POCI rate		48.8%/42.4%/12.8%	

Number (*n*), atrial fibrillation (Af), chronic kidney disease (CKD), mean velocity (MV), peak systolic velocity (PSV), end diastolic velocity (EDV), pulsatility index (PI), computer tomography (CT), magnetic resonance imaging (MRI), total ischemic stroke (TIS), anterior circulation infarction (ACI), posterior circulation infarction (POCI), vertebral artery (VA), and basilar artery (BA), right (RT), left (LT).

**Table 2 jcm-11-01396-t002:** Correlation of sonographic parameters of VAs and BA to age.

Parameters	Pearson’s r (95%CI)/*p* Value	ad-Pearson’s r (95%CI)/ ad-*p* Value
MV of BA	−0.203(−0.272–−0.138)/<0.001	−0.203(−0.247–−0.156)/0.001
PI of BA	0.306(0.240–0.364)/<0.001	0.306(0.269–0.343)/<0.001
MV of RT VA	−0.178(−0.240–−0.115)<0.001	−0.178(−0.228–−0.131)/0.032
PI of RT VA	0.309(0.250–0.370)/<0.001	0.309(0.271–0.343)/<0.001
MV of LT VA	−0.199(−0.259–−0.135)/<0.001	−0.199(−0.237–−0.162)/0.003
PI of LT VA	0.299(0.228–0.369)/<0.001	0.299(0.259–0.339)/<0.001

Mean velocity (MV), pulsatility index (PI), vertebral artery (VA), basilar artery (BA), right (RT), left (LT), confidence interval (CI), and adjusted (ad).

**Table 3 jcm-11-01396-t003:** Adjusted odds ratio and 95% confidence interval (CI) according to grouped analysis of different velocities.

	TIS Rate AOR (95% CI)	POCI Rate AOR (95% CI)
MV ≤ 15 cm/s	1.75 (1.15–2.67) *p* value = 0.009	2.55 (1.58–4.13) *p* value < 0.001
MV ≤ 20 cm/s	1.62 (1.21–2.17) *p* value = 0.001	1.75 (1.15–2.66) *p* value = 0.009
MV ≤ 25 cm/s	1.31 (0.96–1.79) *p* value = 0.094	1.27 (0.79–2.04) *p* value = 0.313
MV ≤ 30 cm/s	0.93 (0.62–1.40) *p* value = 0.728	1.43 (0.75–2.75) *p* value = 0.278
MV ≥ 55 cm/s	1.41 (0.87–2.27) *p* value = 0.16	0.99 (0.51–1.96) *p* value = 0.985
MV ≥ 60 cm/s	2.57 (1.36–4.86) *p* value = 0.004	1.43 (0.67–3.06) *p* value = 0.357

Mean velocity (MV), total ischemic stroke (TIS), posterior circulation infarction (POCI), adjusted odds ratio (AOR), and confidence interval (CI).

## Data Availability

The data presented in this study are available on request from the corresponding author.

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
