# Peer review of "Low Vertebrobasilar Velocity Is Associated with a Higher Risk of Posterior Circulation Ischemic Lesions"

_jcm, 2022, doi:10.3390/jcm11051396_

Round 1

Reviewer 1 Report

Referee Report

MDPI Journal of Clinical Medicine

jcm-1603182

"Low vertebrobasilar velocity is associated with a higher risk of 2 posterior circulation infarction"

Authors retrospectively reviewed records of patients who were suspected of having vertebrobasilar insufficiency (VBI). A total of 875 patients were included in this retrospective study. Statistical analyses were performed with SPSS software version 17.0. Below are a few comments listed. 

[1] Authors applied a multinomial logistic regression was performed to analyze stroke risk among the different groups, expressed as adjusted odds ratios. The validity of the results depends on whether the assumptions of logistic regression models or generalized linear models are satisfied or not. Therefore, it is necessary to perform regression diagnostics. If some assumptions are violated, authors should consider using other models instead.

[2] Authors reported characteristics of patients in Table 1. Besides the column of overall summary statistics, authors should include additional columns for posterior circulation infarction (PCI) group, total ischemic stroke (TIS) group, and the group of others.

[3] For table 1, authors should provide adjusted p-values as the last column, which helps give reader a direct impression on the difference among all groups.

[4] For Table 2. Correlation of sonographic parameters of VAs and BA to age, besides the p-values, authors should compute the confidence intervals for all Pearson’s correlation coefficients.

[5] For the correlation coefficients reported in Table 2, authors need to consider the multiplicity issue. For example, authors computed multiple confidence intervals simultaneously, or compute multiple p-values simultaneously. Authors can provide both raw confidence intervals/p-values and confidence intervals/p-values adjusted for multiplicity. Authors may refer to "Multiple Comparison Procedures" by Hochberg and Tamhane (1987). 

[6] Minor: Extra space between line 79 and 80 on page 2 should be removed.   

Author Response

[1] Authors applied a multinomial logistic regression was performed to analyze stroke risk among the different groups, expressed as adjusted odds ratios. The validity of the results depends on whether the assumptions of logistic regression models or generalized linear models are satisfied or not. Therefore, it is necessary to perform regression diagnostics. If some assumptions are violated, authors should consider using other models instead.

A: Thank you for underlying this deficiency. We performed regression diagnostic test , all variance inflation factor (VIF) values were ≤ 1.2 (no multicollinearity among predictors), and the predictor age was linearly related to the log odds (Box-Tidwell approach)

[2] Authors reported characteristics of patients in Table 1. Besides the column of overall summary statistics, authors should include additional columns for posterior circulation infarction (PCI) group, total ischemic stroke (TIS) group, and the group of others.

 A: Thank you for this useful comment. We added this portion, PCI/TIS groups in Table 1 (due to the problems of composing, we did not add the columns of ACI). 

[3] For table 1, authors should provide adjusted p-values as the last column, which helps give reader a direct impression on the difference among all groups.

 A: Thank you for this useful comment. In the table 1, we compared the basic characteristics and adjusted p-value on the PCI and TIS groups in table 1.

[4] For Table 2. Correlation of sonographic parameters of VAs and BA to age, besides the p-values, authors should compute the confidence intervals for all Pearson’s correlation coefficients.

  A: Thank you for this useful comment. We adjusted this portion in Table 2

[5] For the correlation coefficients reported in Table 2, authors need to consider the multiplicity issue. For example, authors computed multiple confidence intervals simultaneously, or compute multiple p-values simultaneously. Authors can provide both raw confidence intervals/p-values and confidence intervals/p-values adjusted for multiplicity. Authors may refer to "Multiple Comparison Procedures" by Hochberg and Tamhane (1987). 

 A: Thank you for this useful comment. We adjusted this portion in Table 2 

[6] Minor: Extra space between line 79 and 80 on page 2 should be removed.

   A: Thank you for this comment. We had corrected this error.

Reviewer 2 Report

The authors evaluated transcranial color-coded sonography and its relationship between ischemic lesions detected by the imaging findings. This study needs to be brushed up to be published in the academic journal. 

First of all, the title is misleading. The word "posterior circulation infarction" sounds like acute ischemic stroke, but it actually means ischemic lesions in the imaging findings. 

Using CT to assess infarction, especially in the brainstem, is fruitless. Patients who only had CT should be excluded from the analysis.

Nonstandard abbreviations confuse readers. According to the author's criteria, PCI + ACI = TIS? The word PCI sounds like coronary intervention and is too famous to be used in other meanings. 

How do the authors define infarction and white matter hyperintensity? 

Are there any patients with a history of cerebrovascular diseases such as ischemic stroke or coronary artery disease? 

Use "sex" for biological sex and "gender" for social gender.

Why VA flow velocity correlates to anterior infarction is poorly explained. To show the usefulness of the VA flow velocity, it should be compared with the flow velocity of the vessels in the anterior circulation (MCA).

The discussion contains too many paragraphs with a small number of sentences. The authors should rewrite them to have appropriate length and a clear message written as the topic sentence (https://examples.yourdictionary.com/examples-of-topic-sentences.html). 

Author Response

The authors evaluated transcranial color-coded sonography and its relationship between ischemic lesions detected by the imaging findings. This study needs to be brushed up to be published in the academic journal. 

First of all, the title is misleading. The word "posterior circulation infarction" sounds like acute ischemic stroke, but it actually means ischemic lesions in the imaging findings. 

A: Thank you for underlining our deficiencies. We completely agreed with your command. When coming to the brain images interpretation in patient with stroke, arterial more specifically, ischemic lesion is the way we describe on image related to infarction in certain arterial territory. With no any purpose of misleading, the title of our article tried to catch an eye on reader to highlight not only higher but also lower vertebrobasilar velocity pose a risk on posterior circulation stroke in patient’s lifetime.

Using CT to assess infarction, especially in the brainstem, is fruitless. Patients who only had CT should be excluded from the analysis.

A: Thank you for underlining this important defect. We did understand the limit sensitivity of CT in small brainstem infarction and we also mentioned it in our discussion portion . But after thorough deliberation, we still tried to include these patients in this study.

The reasons are as follows:  

  1. The initial proposal of our study was tried to recognize the risk of stroke among clinically diagnosed VBI patient. More than 70% of candidates in our study was coming from out-patient clinic. Patients who walk in the clinic with unspecific in-equilibrium or subtle focal neurologic signs usually received sonography examinations first or brain images according to the clinical scenario. Due to the accessibility and affordability of brain image study, brain CT is always the first option on evaluating outpatient at current medical setting. Therefore, in conjunction to sonography checkup most of patient receive brain CT study only (53.1% in this study).

  1. For those whom underwent brain MRI in our study, most of them were inpatients with older age and more complicated underlying diseases. Inevitably, selection bias in patient group distribution presented in the study. For those reasons, we deeply apologized for our decisions and thanks again for your wise and farsighted comments. With your useful reminding, we are thinking of conducting another following study to confirm this study, focusing on inpatients groups with both MRI and sonography examinations. .

Nonstandard abbreviations confuse readers. According to the author's criteria, PCI + ACI = TIS? The word PCI sounds like coronary intervention and is too famous to be used in other meanings. 

A: Thank you for underlining these deficiencies. We corrected our abbreviations as following:.  Posterior circulation infarction (POCI) abbreviation In this study, we did define that POCI+ACI = TIS.

How do the authors define infarction and white matter hyperintensity? 

A: Thank you for this important question. In this study, this portion CT and MRI images evaluated by one well experienced neuroradiologist by the identical standard. According to the statements of our neuro-radiologist, Dr. Huang, the lesions met with the following characteristics would be diagnosed as white matter hyperintensity, including ill-defined margins, patchy or diffuse appearance in subcortical area, located at specific region (such as periventricular with or without extending to centrum semiovale), with halos or multiple punctuate lesions. Sometimes, it is difficulty to differentiate white matter hyperintensity and small subcortical lacunes indeed. But lacune lesions tend to scattered or isolated distribution with clear margin and containing small CSF cavies  

Are there any patients with a history of cerebrovascular diseases such as ischemic stroke or coronary artery disease? 

A: Thank you for underlining these deficiencies. In this study, some of them with a history of ischemic stroke or coronary artery disease; however, due to incomplete recordings of these risks, we did not present this data.

Use "sex" for biological sex and "gender" for social gender.

A: Thank you for underlining these deficiencies; we have corrected this error.

Why VA flow velocity correlates to anterior infarction is poorly explained. To show the usefulness of the VA flow velocity, it should be compared with the flow velocity of the vessels in the anterior circulation (MCA).

A: Thank you for this useful comment. Generally, the vascular condition in the anterior circulation would correlate to the flow findings in MCA/ACA mostly. Regarding to the relationship between VA flow velocity and anterior circulation stroke observed in our study, we speculated possible conditions as generalized atherosclerosis, aging and inadequate systemic circulation related to heart problems may play a role on it.  Owing to demographic differences, elderly and female individuals have higher failure rates of performing sonography over the temporal window. Conversely, there is no such limitation to approach vertebrobasilar system from occipital window. Therefore, the clinical value of abnormalities in VB flow not only associate with higher posterior circulation strokes but also alarm the possibility of concurrent abnormalities in rest part of intracranial system.

The discussion contains too many paragraphs with a small number of sentences. The authors should rewrite them to have appropriate length and a clear message written as the topic sentence (https://examples.yourdictionary.com/examples-of-topic-sentences.html). 

A: Thank you for underlining these deficiencies. We have rewritten our discussion.

Reviewer 3 Report

Nice job. I have some questions:

  • the same radiologist/neurologist made transcranial color-coded sonography?
  • the same radiologist/neurologist evaluated CT and MRI images? what  is the experience of the radiologist?
  • was the CT and MRI images evaluated "double blind'? 
  • all patients passed CT and MRI exam? (as you know the CT exam some time have negative evaluation and after the MRI exam we may find stroke in the vertebral arteries)

Author Response

  • the same radiologist/neurologist made transcranial color-coded sonography?

A: Thanks for this important comment. All of the transcranial color-coded sonography evaluation in our case was done by well-trained and experienced sonography technicians, under the same machine setting and regular protocols of measuring target vessels. However, limitation as the determination of VB velocity is operator dependent with a difficulty reducing the technique error during examination.

  • the same radiologist/neurologist evaluated CT and MRI images? what  is the experience of the radiologist?

A: Thanks for this important question. For brain CT and MRI images interpretation, it was reviewed and confirmed by the same well experienced neuroradiologists who taking in charge of the neuroimage reports and procedures in our hospital for more than 20 years.

  • was the CT and MRI images evaluated "double blind'? 

A: Thanks for this important question.  Neither brain CT nor MRI performance was double-blinded, regarding to the retrospective study design and patient selection based on the clinical suspicion of VBI.

  • all patients passed CT and MRI exam? (as you know the CT exam some time have negative evaluation and after the MRI exam we may find stroke in the vertebral arteries)

A: Thanks for this important question. Comparing to the accessibility and affordability of brain image study, brain CT is always the first option on evaluating patient with acute or subacute focal neurologic sign at current medical setting. However, even for one experienced neuroradiologist, it is always a challenge to confirm posterior circulation stroke on brain CT, particularly in patients with subtle symptoms due to anatomical complexity and possibility of skull base artifacts that might mask posterior circulation lesions. And this corresponded to most of our PCI infarcts was confirmed by MRI (89.3%), and we could not deny the possibility of omitting PCI lesions.

Round 2

Reviewer 1 Report

Authors have addressed all my comments and I have no further comment. 

Author Response

Thank you for the time and useful comments to improve this manuscript.

Reviewer 2 Report

The authors stated the title is determined to be catchy and could not be changed. However, using a misleading title is unfaithful. The title should be a brief summary of the research itself. "Ischemic lesion" is better than "infarction". 

Also, the authors described in the conclusion that "lower VB velocity may be associated with higher rates of both anterior and posterior infarctions". If they think the correlation between VB velocity and both anterior and posterior infarction is significant, it should be included in the title. 

The authors should include the reason of including patients with CT only and how they define infarction and white matter hyperintensity in the main text. 

The definition of POCI, AIS and TIS should be more clarified and it is better to use similar abbreviations (e.g. POCI, AOCI and TOCI  or PIS, AIS and TIS). It should be clearly stated that  the sum of "posterior" and "anterior" makes "total".  

Author Response

The authors stated the title is determined to be catchy and could not be changed. However, using a misleading title is unfaithful. The title should be a brief summary of the research itself. "Ischemic lesion" is better than "infarction". 

A: Thank you for underlining these deficiencies and from your feedback, we appreciate the importance and responsibility to express the idea of title precisely. Therefore, we corrected the diction error and changed the title as,

“Low vertebrobasilar velocity is associated with a higher risk of posterior circulation ischemic lesions”

Thank you again for the reminding, and we will be more careful on handling word selection in the future.

Also, the authors described in the conclusion that "lower VB velocity may be associated with higher rates of both anterior and posterior infarctions". If they think the correlation between VB velocity and both anterior and posterior infarction is significant, it should be included in the title. 

A: Thank you for the comment and highlight the potential strength of our article. The initial proposal of our study was tried to recognize the risk of stroke among clinically diagnosed VBI patients; therefore, the primary goal was aiming on the association between symptoms and sonographic parameters in the posterior circulation. However, the result of sonographic findings and demographic data gave as an additional observation of higher risk of the anterior infarction in the group of lowered VB flow; we take it as a valuable finding. With the inspiration, we hope to conduct another following study to confirm the additional objective finding, focusing on patients with anterior circulation syndromes and sonographic findings. In order to stay with the primary goal of our study, may allow us to keep our title “Low vertebrobasilar velocity is associated with a higher risk of posterior circulation ischemic lesions”. 

The authors should include the reason of including patients with CT only and how they define infarction and white matter hyperintensity in the main text. 

A: Thank you for this useful comment. We added and addressed these contents in the part of methods (Brain images and criteria of stroke) and discussion.

The definition of POCI, AIS and TIS should be more clarified and it is better to use similar abbreviations (e.g. POCI, AOCI and TOCI  or PIS, AIS and TIS). It should be clearly stated that  the sum of "posterior" and "anterior" makes "total".  

 A: Thank you for this useful comment. After discussion and literature reviewing, we decided to adapt the commonest abbreviations used in medical reports (POCI, ACI and TIS) and added the definitions in methods (Brain images and criteria of stroke).